# Non-enzymatic glycolysis and pentose phosphate pathway-like reactions in a plausible Archean ocean

Markus A Keller[1], Alexandra V Turchyn[2] & Markus Ralser[1,3,*]

## Abstract

The reaction sequences of central metabolism, glycolysis and the pentose phosphate pathway provide essential precursors for nucleic acids, amino acids and lipids. However, their evolutionary origins are not yet understood. Here, we provide evidence that their structure could have been fundamentally shaped by the general chemical environments in earth's earliest oceans. We reconstructed potential scenarios for oceans of the prebiotic Archean based on the composition of early sediments. We report that the resultant reaction milieu catalyses the interconversion of metabolites that in modern organisms constitute glycolysis and the pentose phosphate pathway. The 29 observed reactions include the formation and/or interconversion of glucose, pyruvate, the nucleic acid precursor ribose-5-phosphate and the amino acid precursor erythrose-4-phosphate, antedating reactions sequences similar to that used by the metabolic pathways. Moreover, the Archean ocean mimetic increased the stability of the phosphorylated intermediates and accelerated the rate of intermediate reactions and pyruvate production. The catalytic capacity of the reconstructed ocean milieu was attributable to its metal content. The reactions were particularly sensitive to ferrous iron Fe(II), which is understood to have had high concentrations in the Archean oceans. These observations reveal that reaction sequences that constitute central carbon metabolism could have been constrained by the iron-rich oceanic environment of the early Archean. The origin of metabolism could thus date back to the prebiotic world.

**Keywords** Archean ocean; evolution of metabolism; glycolysis; non-enzymatic catalysts; pentose phosphate pathway

**Subject Categories** Evolution; Metabolism

**Mol Syst Biol. (2014) 10: 725**

See also: **PL Luisi** (April 2014)

## Introduction

In the early 20th century, Aleksandr I Oparin formulated a hypothesis that the origin of life could have been facilitated by the geochemical formation of cellular constituents in the reducing environment of the early earth (Oparin, 1938; Miller *et al*, 1997). Experimental support for this hypothesis was provided by the Miller–Urey experiment that demonstrated the non-biological and simultaneous synthesis of amino acids upon replicating a hypothesized early–earth atmosphere. Miller had combined water ($H_2O$), methane ($CH_4$), ammonia ($NH_3$) and hydrogen ($H_2$), which resulted in the enzyme-free formation of alanine and glycine upon applying energy in the form of high-voltage electrical discharges (Miller, 1953). A recent re-examination of samples from a later experiment containing hydrogen sulphide demonstrated the formation of virtually all proteogenic amino acids (Parker *et al*, 2011). The results by Miller stimulated series of further studies that demonstrated the formation of a multitude of bio-essential molecules in geochemical reactions and catalysed by minerals and on mineral surfaces, metals and other chemical components, which could have been present on the early world. The observed reactions include the generation of potential precursors for RNA, presumed to be a key molecule in early evolution (Wachtershauser, 1988; Wächtershäuser, 1990; Orgel, 2004; Powner *et al*, 2009; Hud *et al*, 2013).

In this study, we address a potential scenario for one of the critical subsequent events, the origin of cellular metabolism. Despite cellular organisms possess sophisticated metabolic capacities, the origin of the underlying metabolic network is only barely understood (Wächtershäuser, 1988; Lazcano & Miller, 1999; Luisi, 2012). Obtaining a metabolic network facilitated by cells, or the ecosystem cycles they were involved in, to self-synthesize their required building blocks (autotrophy) will render them independent to specific geological conditions. This is an essential feature to colonize different niches on the earth and to persist over geological time-frames.

As the core structure of the metabolic network is similar in all organisms examined (Jeong *et al*, 2000; Braakman & Smith, 2013), one can assume that the metabolic network came into existence at the very early stages of evolution. In principle, two scenarios are

1 Department of Biochemistry and Cambridge Systems Biology Centre, University of Cambridge, Cambridge, UK
2 Department of Earth Sciences, University of Cambridge, Cambridge, UK
3 Division of Physiology and Metabolism, MRC National Institute for Medical Research, Mill Hill, London, UK
*Corresponding author. Tel: +44 1223 761346; Fax: +44 1223 766002; E-mail: mr559@cam.ac.uk

considered: (i) modern biochemical reaction sequences are the result of evolutionary selection. This hypothesis would dictate that the structure of the modern metabolic network differs to that used by the most primitive organisms (Lazcano & Miller, 1999; Anet, 2004). The alternative scenario (ii) is that the initial metabolic reaction network was pre-established on the prebiotic earth and was consequence of the chemical and physical environment when life first emerged. These chemically constrained reaction sequences would then be incorporated into cells; the evolution of enzymes would have occurred later and accelerated the reactions, increasing the specificity and extending this original network. If this latter hypothesis is correct, the core reaction sequences of modern metabolism today could still resemble the one used by the first living organisms (Wächtershäuser, 1988, 1990; Shapiro, 2000; Huber *et al*, 2012). Overall, there are arguments supporting both of these hypotheses (Luisi, 2012; Braakman & Smith, 2013), but firm experimental evidence for either scenario is missing.

Due to the complexity of the metabolic pathways, it has been argued that metabolism-like chemical reaction sequences are unlikely to be catalysed by simple environmental catalysts (Lazcano & Miller, 1999; Anet, 2004). However, to our knowledge, this possibility has not been tested systematically, and at present stage, thermodynamic approaches are not predictive about which molecules form in the presence of simple catalysts from a precursor (Amend *et al*, 2013). Here, we studied the reactivity of intermediates of glycolysis and the pentose phosphate pathway upon replicating a plausible chemical composition of the prebiotic Archean ocean. We report that under these conditions, the intermediates of the two pathways undergo non-enzymatic interconversion reactions and form neighbouring intermediates that constitute these pathways in modern cells. Several of the observed reactions replicate enzyme-catalysed reactions. In addition, we find that the conditions that replicate an average ionic oceanic environment of the Archean favour the stability of sugar phosphate metabolites, increase the specificity of non-enzymatic pyruvate formation and accelerate glycolysis- and pentose phosphate-like non-enzymatic reactions. These results therefore favour a hypothesis that abundant ions of the Archean ocean could have served as catalysts for the first forms of metabolism. The origin of central carbon metabolism would thus date back to chemical constraints of the prebiotic ocean.

## Results

### Spontaneous reactions of glycolytic and the pentose phosphate pathway metabolites form metabolic intermediates

The biosynthetic machinery of the modern cell is dependent on a series of sugar phosphate metabolites. Known as glycolysis (Embden-Meyerhof-Parnass pathway) and the pentose phosphate pathway, sugar phosphates metabolites are interconverted through common and conserved reaction sequences. Glycolysis and the pentose phosphate pathway overlap and share most reactions with the Entner-Doudoroff pathway and the Calvin cycle. The Entner-Doudoroff pathway is common in bacteria and is specific for the metabolite 2-keto-3-deoxyphosphogluconate, whereas the Calvin cycle is common in photosynthetic organisms. While the reactions that constitute these pathways are enzyme-catalysed in modern

organisms, in some cases, non-enzymatic pendants have been observed. For instance, Reynolds *et al* described the equilibrium between the three-carbon sugar phosphates dihydroxyacetone phosphate (DHAP) and glyceraldehyde 3-phosphate (G3P), in modern metabolism converted by triosephosphate isomerase (Reynolds *et al*, 1971). More recently, Breslow presented a chemical mimetic for transaldolase (Breslow & Appayee, 2013), indicating that the enzyme-catalysed reactions of modern metabolism can also occur spontaneously, or can be catalysed using much simpler, chemical catalysts.

We used liquid chromatography-selective reaction monitoring (LC-SRM) to systematically elucidate non-enzymatically catalysed interconversions of metabolites that constitute glycolysis and the pentose phosphate pathway. The LC-SRM technique facilitates such an investigation, as its data-independent acquisition nature allows not only conclusions about the presence, but also about the absence of a given molecule (considering the specific limits of detection, which on modern instruments is typically at femtomoles, rendering LC-SRM the to date most sensitive LC-MS/MS technique), and as it yields absolute quantities, enables calculation of reaction rates. Most importantly, however, the high sensitivity of the targeted method facilitates full coverage of all intermediates of a given pathway at a reasonable throughput, so that reaction conditions can be systematically tested.

We first determined the spontaneous reactivity upon heat exposure of the metabolic intermediates glucose 6-phosphate (G6P), fructose 6-phosphate (F6P), fructose 1,6-bisphosphate (F16BP), dihydroxyacetone phosphate (DHAP), glyceraldehyde 3-phosphate (G3P), 3-phosphoglycerate (3PG), phosphoenolpyruvate (PEP), 6-phosphogluconate (6PG), ribulose 5-phosphate (Ru5P), ribose 5-phosphate (R5P), xylulose 5-phosphate (X5P) and sedoheptulose 7-phosphate (S7P) (Fig 1). The chemicals were dissolved each at 100 μM concentration, a concentration that is lower than their intracellular level (Wamelink *et al*, 2005; Bennett *et al*, 2009) in high-purity ULC-MS grade water. The sugar phosphate solutions, containing the metabolites separately, were incubated for 5 h at 70°C. This temperature is plausible for oceanic environments close to heat sources such as hydrothermal vents of volcanoes, but substantially exceeds the heat stability of typical metabolic enzymes. The samples were then separated on an ultra-high pressure HPLC system (Agilent 1290) coupled to a triple quadrupole mass spectrometer (Agilent 6460) operating in SRM mode. The identity of each metabolites was assessed by matching mass (m/z), fragmentation pattern, retention time and co-occurring SRM transitions with chemical metabolite standards, adapting our quantification methods published previously (Wamelink *et al*, 2005; Ralser *et al*, 2009). Using this set-up, absolute quantities were obtained for 15 intermediates of glycolysis and the pentose phosphate pathway, facilitating monitoring 182 possible reactions per experiment. This corresponds to full coverage of the reaction space within the two pathways, except the formation of two intermediates (1,3-bisphosphoglycerate and 6-phosphoglucono-δ-lactone) that were not observed in our set-up. The structural isomers 2PG/3PG and Ru5P/X5P could not be distinguished due to their similar mass, structure and separation properties and were quantified as pools.

In water, we observed 17 cases in which a glycolytic or a pentose phosphate pathway intermediate converted into another intermediate metabolite (Fig 1A, lower panel). In the pentose phosphate

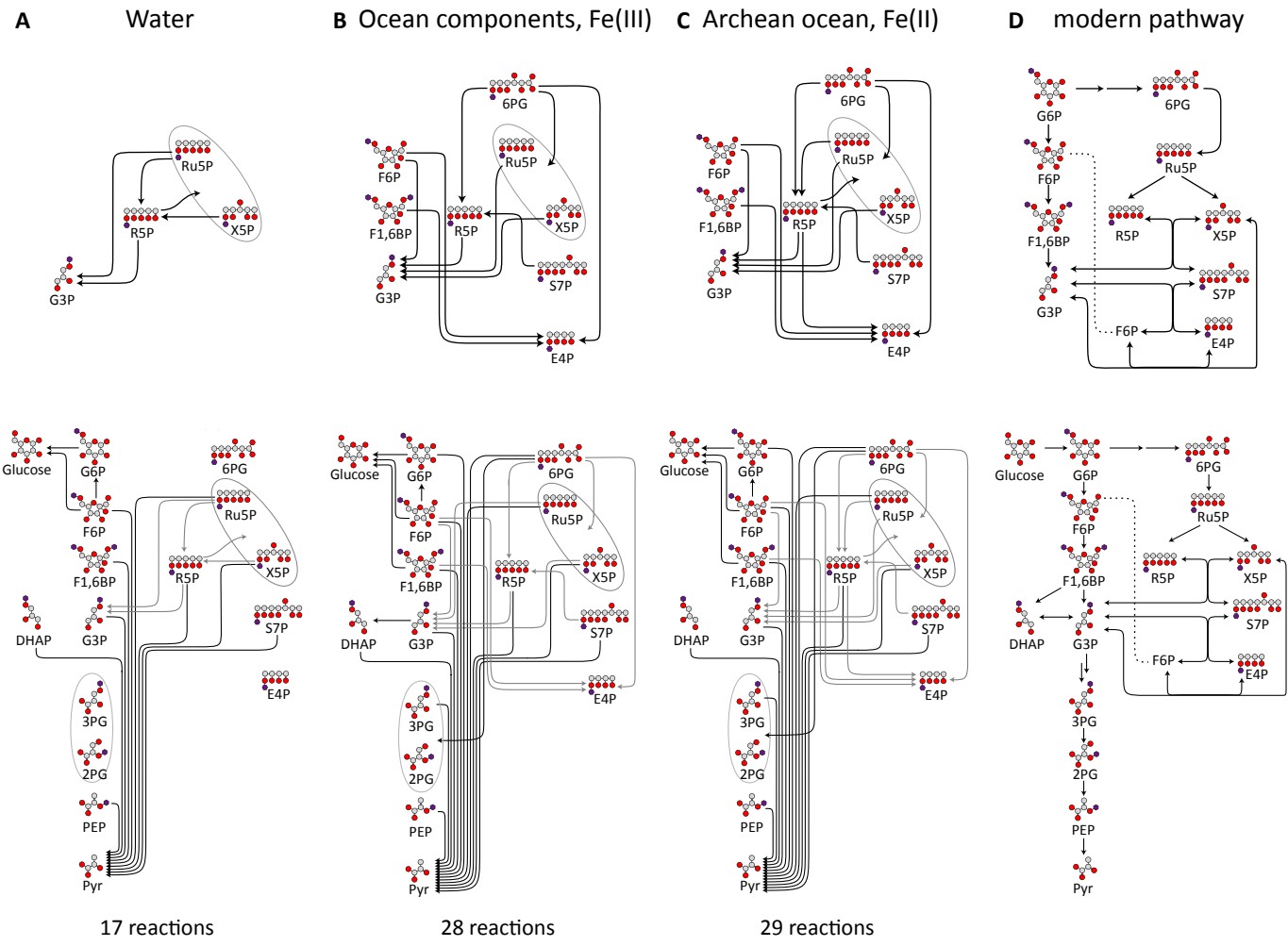

**A** Water   **B** Ocean components, Fe(III)   **C** Archean ocean, Fe(II)   **D** modern pathway

17 reactions        28 reactions        29 reactions

**Figure 1.  Non-enzymatic sugar phosphate interconversions in a plausible Archean ocean environment reproduce reactions from glycolysis and the pentose phosphate pathway (PPP).**

A  Spontaneous reactivity of glycolytic and pentose phosphate pathway sugar phosphate intermediates as observed in water. One hundred micromolar of the intermediates (the upper panels illustrate the pentose phosphate pathway, the lower panels the combined pentose phosphate pathway and glycolysis) was monitored for 5 h at 70°C. Time-dependent accumulation of metabolites was considered significant when observed in at least three independent experiments. Arrows connect the substrate with the detected metabolites formed. Every SRM measurement targeted the full spectra of glycolytic and the PPP metabolites. 2-phosphoglycerate (2PG) and 3-phosphoglycerate (3PG) as well as xylulose 5-phosphate and ribulose 5-phosphate could not be separated due to similar retention time, mass and fragmentation spectra and were quantified in pools (grey circles). Metabolites not quantifiable with the approach were 1,3-bisphosphoglycerate and 6-phosphoglucono-δ-lactone and are omitted from the graph. In total, of the 182 monitored possible reactions, 17 were detected to occur in water. Individual reactions are detailed in Supplementary Table S1; rates are given in Fig 5A.

B  Plausible Archean ocean constituents catalyse metabolism-like reactions. The same metabolites and reactions were monitored as in (A), with the difference that the reactions were conducted in the presence of Archean ocean plausible concentrations of iron, cobalt, nickel, molybdenum and phosphate. In this milieu, 28 interconversion reactions among glycolytic and pentose phosphate pathway intermediates were observed.

C  The influence of ferrous iron [Fe(II)] under anoxic conditions. The same metabolites and reactions were monitored as in (B), with the difference that iron was maintained ferrous Fe(II), replicating the reducing conditions of the early oceans. In this reaction milieu, 29 metabolite formation reactions were detected. Differences to (B) concern additional interconversion of pentose phosphate metabolites, and fewer interconversions of 3-carbon metabolites. Individual reactions are detailed in Supplementary Table S1 and Fig 5A.

D  Network topology of modern glycolysis (canonical Embden-Meyerhof pathway) and the pentose phosphate pathway.

Data information: abbreviations used in (A–C): *Pentose phosphate pathway:* 6PG, 6-phosphogluconate; Ru5P, ribulose 5-phosphate; R5P, ribose 5-phosphate; X5P, xylulose 5-phosphate; S7P, sedoheptulose 7-phosphate; E4P, erythrose 4-phosphate. *Glycolysis:* G6P, glucose 6-phosphate; F6P, fructose 6-phosphate; F16BP, fructose 1,6-bisphosphate; DHAP, dihydroxyacetone phosphate; G3P, glyceraldehyde 3-phosphate; 3PG, 3-phosphoglycerate; 2PG, 2-phosphoglycerate; PEP, phosphoenolpyruvate; Pyr, pyruvate.

pathway, we observed isomerization of ribose 5-phosphate, ribulose 5-phosphate and xylulose 5-phosphate, as well as the formation of glyceraldehyde 3-phosphate from these intermediates (Fig 1A, upper panel). In cellular metabolism, analogous reaction sequences do

exist and are catalysed by the pentose phosphate pathway enzymes ribulose 5-phosphate epimerase, ribose 5-phosphate isomerase and transketolase (Fig 1D, upper panel). Glycolytic intermediates converted into pyruvate and glucose, the stable products of

glycolysis and gluconeogenesis, as well as the intermediate metabolite glucose 6-phosphate (Fig 1A, lower panel). Overall, pyruvate formation dominated. Its non-enzymatic formation was detected from PPP metabolites, fructose 6-phosphate, fructose 1,6-bisphosphate and all intermediates of lower glycolysis (Fig 1D, lower panel). It is therefore apparent that heat exposure is sufficient to convert intermediate metabolites of glycolysis and the pentose phosphate pathway into pyruvate and glucose that constitute thermodynamically stable products also in the modern, enzyme-catalysed metabolism, and to induce isomerization between pentose phosphate metabolites.

### Chemical constituents of the pre-oxygenation oceans catalyse multiple complex sugar phosphate interconversion reactions

We next tested whether conditions that replicate the environment in which first cellular organisms evolved, the pre-oxygenation oceans of the Archean, would have an influence on the sugar phosphate interconversion reactions. From the chemical composition of sediments dated to this period, it has been asserted previously that the Archean ocean was carbon rich and contained significant concentrations of Na, Cl, K, $BO_3$, F, $PO_4$, Mg, Ca, Si, Mo, Co, Ni and Fe. To simulate a reaction environment of the Archean ocean, these molecules were combined at their Archean ocean-plausible concentration ranges (Van Cappellen & Ingall, 1996; Canfield, 2000; Saito *et al*, 2003; Rouxel *et al*, 2005; Zerkle, 2005; Scott *et al*, 2012; Robbins *et al*, 2013) with the metabolic intermediates, and the reactivity was monitored for 5 h at 70°C. Similar to the experiments described above, the LC-SRM method was set to monitor the formation of intermediates of glycolysis and the pentose phosphate pathway. These metabolites were absolutely quantified in the resultant 1,200 samples by manually supervised LC-MS/MS peak identification and integration of the approximately 18,000 resultant chromatographic

peaks (raw data are provided in the Supplementary Table S5). We did not detect a significant influence of the presence or absence of the oceanic salts on the sugar phosphate interconversion reactions. However, a reactive solution containing the metal ions Fe, Co, Ni, Mo as well as phosphate at Archean ocean plausible concentrations catalysed additional reactions. In the metal-rich solution, out of the 182 reactions monitored, we observed in 28 cases the conversion of a glycolytic or pentose phosphate metabolite into another intermediate of the two pathways (Fig 1B). The reactions catalysed by the Archean ocean mimetic yielded the formation of ribose 5-phosphate, the constituent of the RNA backbone and erythrose 4-phosphates, the precursor of aromatic amino acids, starting from several pentose phosphate pathway intermediates. As a result, these non-enzymatic reactions interconnected all pentose phosphate pathway intermediates (glucose 6-phosphate, erythrose 4-phosphate, ribose 5-phosphate, ribulose 5-phosphate, xylulose 5-phosphate and sedoheptulose 7-phosphate). The interconversion network corresponds to the reaction sequence of the modern-cell non-oxidative pentose pathway (Fig 1B and D). In difference, interconversion reactions that would correspond to reactions of the oxidative pentose phosphate pathway, converting glucose 6-phosphate to 6-phosphogluconate, were not observed.

### Iron is the most predominant catalyst in the early Archean ocean simulation

The catalytic potential of the Archean ocean mimetic is exemplified for the central pentose phosphate pathway metabolite 6-phosphogluconate (6PG). 6PG was stable at 70°C in water and did not interconvert in any glycolytic or other pentose phosphate metabolites. However, upon adding the Archean ocean ionic constituents Fe, Ni, Co, Mo and phosphate, this metabolite reacted into intermediates of the pentose phosphate pathway and yielded pyruvate (Fig 2A).

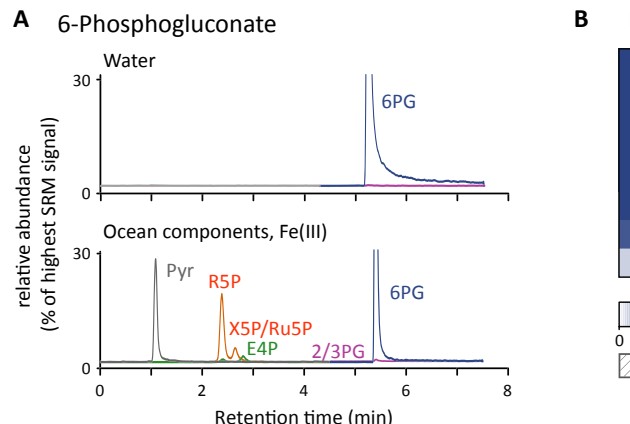

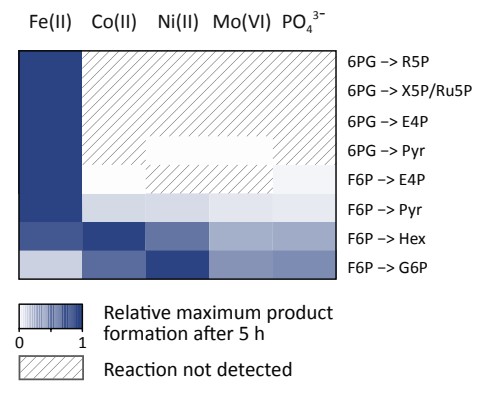

**Figure 2. Catalytic capacity of the Archean ocean constituents.**

A  The Archean ocean ionic composition catalyses sugar phosphate interconversions. 6-phosphogluconate (6PG) was incubated at 70°C in water, or in the presence of Archean ocean plausible concentrations of Fe, Co, Ni, Mo and phosphate. The chromatograms illustrate an exemplary LC-SRM run targeting the glycolytic and pentose phosphate pathway intermediates recorded after 2 h. 6PG was stable in water (upper panel), but was interconverted into other pentose phosphate pathway intermediates and pyruvate as catalysed by the Archean ocean components (lower panel).

B  Iron is the predominant catalyst for pentose phosphate pathway interconversions. 6-phosphogluconate (6PG) and fructose 6-phosphate (F6P) were incubated at 70°C in the presence of the indicated Archean ocean constituents, and the formation of reaction products was monitored by LC-SRM over 2 h. The reaction rates were calculated as detailed in the Materials and Methods section; metabolite concentrations are provided in Supplementary Table S5. Ferrous iron facilitated the interconversion of the metabolites into eight metabolic intermediates, whereas Co, Ni, Mo and phosphate together contributed to a subset of the reactions. Non-detectable reactions are represented by hatched areas.

We next tested the contribution of the individual mimetic constituents to the reactivity. The reactivity of 6-phosphogluconate (6PG) or fructose 6-phosphate (F6P), specifically inter-converted by the Archean ocean mimetic, was mostly sensitive to the presence of iron. The other metal constituents (Co, Ni, Mo), as well as phosphate, contributed to the catalysis of a subset of the interconversion reactions only (Fig 2B). Indeed, iron is of particular importance in the Archean ocean, as its concentration was likely much higher than today. This is the consequence of the different solubility of ferric and ferrous iron: whereas ferrous iron is readily water soluble in the absence of oxygen, its oxidized form Fe(III) is not. Indeed, the geochemical record indicates that anoxic conditions were present over most of early earth's surface environment. This assumption is supported by the presence of iron formations, detrital uraninite and pyrite, the preserved mass-independent fractionation of sulphur isotopes and the lack of cerium anomalies in paleosols, whose formation requires the absence of free oxygen. It is thus assumed that the Archean milieu preserved a high iron content, whereas its oxidized form was then readily depleted upon the great oxygenation event (Holland, 1984, 2006; MacFarlane *et al*, 1994; Farquhar, 2000; Kasting & Ono, 2006).

To simulate an Archean ocean in the presence of ferrous iron [Fe(II)], the following series of experiments were conducted under anoxic conditions by controlling the $O_2$ concentration below 8 ppm, and monitoring the iron redox state (Materials and Methods). Whereas iron was present as ferric iron in the normoxic solutions, the anoxic conditions preserved iron in its ferrous form. Monitoring 182 possible interconversion reactions in the ferrous iron-rich mimetic, we detected the formation of glycolytic or pentose phosphate pathway metabolites in 29 cases. Compared to the analogous Fe(III)-rich solution, most additional interconversion reactions concerned pentose phosphate intermediates. The additional reactions facilitated the formation and interconversion of ribose 5-phosphate and erythrose 4-phosphate, while interconversions of some three-carbon phosphates were lost (Fig 1C, upper panel). The additional reactions observed in the presence of Fe(II) did not trigger a major topological change compared to the interconversion network observed in the presence of Fe(III) (Fig 1B and C). However, as shown below, the presence of ferrous iron had major effects on metabolite stability and increased reaction rates as well as the efficiency of the interconversion reactions.

### Glycolytic and PPP intermediate interconversions gain specificity in an Fe(II)-rich anoxic Archean ocean mimetic

To address the stability of the glycolytic and pentose phosphate pathway intermediates, we analysed reactivity in a time-dependent manner. A time series upon heat exposure is exemplified for glyceraldehyde 3-phosphate (G3P), the least stable glycolytic intermediate. At 70°C and in water, this metabolite degrades within minutes (Fig 3A). Upon addition of the Archean oceanic chemical constituents, 50% of G3P was still present after 30 min. In the reaction mixture containing ferrous iron, G3P was further stabilized and could be detected after 5 h and longer (Fig 3A). Such time-series experiments were subsequently conducted for all glycolytic and pentose phosphate pathway intermediates. The G3P result was representative for all phosphorylated intermediates: the constituents gained stability in the presence of the Archean ocean solution containing ferrous iron relative to the ferric conditions. All sugar phosphate intermediates increased in stability up to 15-fold (Fig 3C).

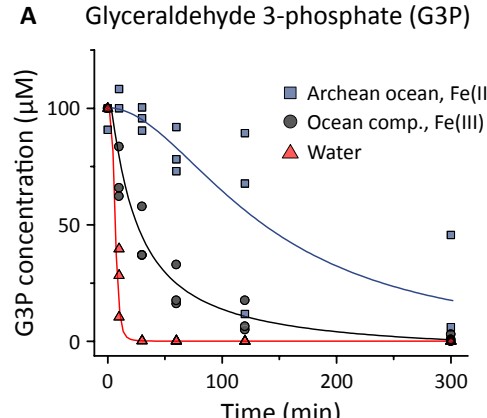
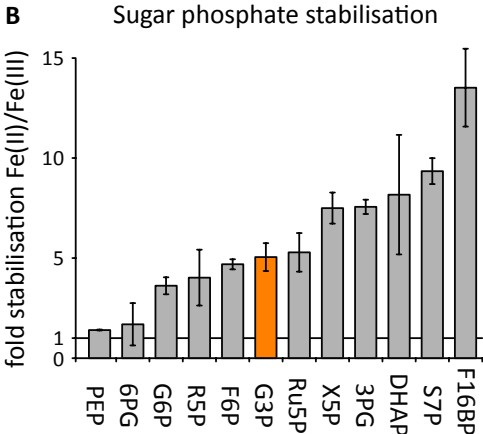

**Figure 3.  The ionic composition of the anoxic Archean increases the stability of metabolic pathway intermediates.**

A  The stability of glyceraldehyde 3-phosphate (G3P) in Archean ocean simulations. G3P was diluted in water, or the Archean ocean mimetic containing Fe(III), Co, Ni, Mo and phosphate, or the analogous anoxic solution containing Fe(II). The solutions exposed to 70°C and monitored by LC-SRM for 5 h. G3P was degraded in water within minutes, was stabilized by the oxygenated, metal-rich ocean mimetic and remained detectable for more than 5 h in the ferrous iron-rich ocean simulation.

B  The ferrous iron-rich Archean ocean ionic composition favours stability of sugar phosphate intermediates. Glycolytic and pentose phosphate pathway intermediates were exposed to 70°C as in (A) and their concentration monitored over 5 h. Illustrated is the fold increase in stability in the Fe(II)-rich Archean ocean mimetic over the corresponding stability in the Fe(III)-rich isoionic solution. All sugar phosphate intermediates that constitute the PPP and glycolysis gained stability. The data obtained in water (Supplementary Tables S2 and S5) are not directly comparable due to the absence of the majority of the pentose phosphate interconversion reactions (Fig 1A and D). Quantitative stability data are given in Supplementary Table S2 and metabolite concentrations in Supplementary Table S5. Error bars indicate ± SD calculated with error propagation.

As the increased retention of the monitored metabolites implied a decrease in non-specific reactivity, we questioned whether the presence of ferrous iron increased specificity in the catalysis of the interconversion reactions. Indeed, ferrous iron is an active, but also specific, catalyst and electron donor frequently used in organic carbohydrate chemistry (Wehrli *et al*, 1989; Casey & Guan, 2007). Absolute sugar phosphate concentrations measured in time-series experiments were used to calculate the specificity of the interconversion reactions. After a 5 h exposure at 70 C˚ in the ferric iron-rich normoxic solution, on average 36.7% of the carbon atoms were present as substrate and 12.3% as newly formed intermediates (Fig 4). Hence, 49% of the carbon was recovered as glycolytic and pentose phosphate pathway intermediates, and the remaining molecules were interconverted into other metabolites. In the presence of ferrous iron, however, specificity increased and a mean carbon recovery of 61.8% was achieved (49.9% as substrates, 11.9% as newly formed intermediates) (Fig 4). Thus, one-fourth of the absolute concentrations of metabolites that were interconverted non-enzymatically into a new molecule formed an intermediate of glycolysis and the pentose phosphate pathway. Considering the large number of carbohydrate compounds that can form in theory from sugar phosphate precursors, and in comparison with typical yields in multi-step organic synthesis processes, indicates that glycolytic and PPP intermediates are favourably formed.

**The Archean ocean ionic composition accelerates the non-enzymatic formation of pyruvate**

In order to test the influence of the increased specificity on the reaction rates, time series were recorded for all metabolic

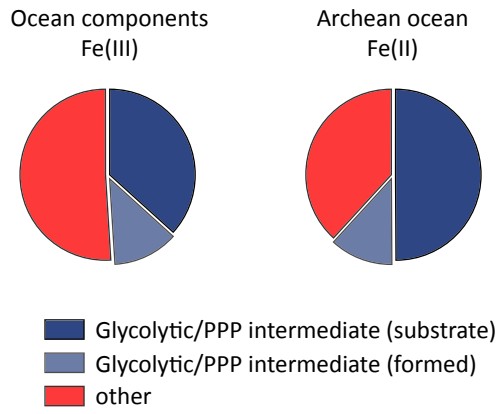

Ocean components Fe(III)          Archean ocean Fe(II)

■ Glycolytic/PPP intermediate (substrate)
■ Glycolytic/PPP intermediate (formed)
■ other

**Figure 4. Efficiency in the formation of glycolytic and PPP intermediates.**
Carbon recovery. Glucose 6-phosphate (G6P), fructose 6-phosphate (F6P), fructose 1,6-bisphosphate (F16BP), dihydroxyacetone phosphate (DHAP), glyceraldehyde 3-phosphate (G3P), 3-phosphoglycerate (3PG), phosphoenolpyruvate (PEP), 6-phosphogluconate (6PG), ribulose 5-phosphate (Ru5P), ribose 5-phosphate (R5P), xylulose 5-phosphate (X5P) and sedoheptulose 7-phosphate (S7P) were exposed for 5 h to 70°C in water and in the Archean ocean chemical mimetics. Absolute quantities obtained by LC-SRM were then used to assess how much carbon was present as the substrates or as newly formed metabolic intermediates. In the presence of Fe(III), 49.9% of carbon was recovered as glycolytic and pentose phosphate pathway metabolites, 61.8% in the presence of Fe(II). In the presence of Fe(III), thus 12.3% of totally formed metabolites were intermediates of glycolysis and the pentose phosphate pathway, 11.9% in the presence of Fe(II).

interconversions detected in the Fe(II)-rich Archean ocean mimetic. The reaction rates were assessed by fitting the detected concentration changes to least-squared linear, nonlinear formation or exponential functions, depending on which best described the observed formation dynamics (Supplementary Table S1). The determined reaction rates ranged over four orders of magnitude (Fig 5A). In the ferrous iron-rich Archean ocean mimetic, the fastest observed reaction occurred at a rate 19.6 µM/h and was the conversion 6-phosphogluconate to the RNA backbone precursor ribose 5-phosphate. In contrast, the slowest reaction, the dephosphorylation of glucose 6-phosphate to glucose, occurred at a rate of 0.09 µM/h (Fig 5A). In order to facilitate a comparison of the individual reaction rates between the different conditions, we normalized every reaction to its maximally observed rate and sorted them using hierarchical clustering (Fig 5B). Several of the interconversion rates were faster in the Archean ocean mimetics compared to the water (12 out of the 17 reactions detected in water); comparing the mimetics, 12 reactions were accelerated in the presence of Fe(II) over Fe(III) (Fig 5B). 6-phosphogluconate for instance was not converted to ribose 5-phosphate in water. In the ferric iron-rich mimetic, this reaction was observed at a rate of 5.5 µM/h, while in the ferrous iron-rich anoxic experiment at a rate of 19.6 µM/h (Fig 5A and B, Supplementary Table S1). In some cases, acceleration was also observed for the three-carbon phosphate interconversion reactions that are analogous to reactions of lower glycolysis; that is, the conversion of phosphoenolpyruvate to pyruvate occurred at 13.5 µM/h in the Fe(II)-rich Archean ocean mimetic, while a rate of 7.3 µM/h was observed in water. However, several of the of three-carbon metabolite formation reactions, including the formation of pyruvate from fructose 1,6-bisphosphate, or glyceraldehyde 3-phosphate from ribose 5-phosphate, were slowed down, most likely due to the stabilization of precursors (Fig 5B).

In order to get a global picture of the reactivity in the system, we therefore combined all intermediate metabolites at a concentration of 7.5 µM and determined the temperature-dependent rate of pyruvate formation. At temperatures of 40°C and below, no pyruvate was formed, confirming the absence of enzymatic contamination in the combined reaction mimetic. Above 50°C, pyruvate formation was detected and increased in a temperature-dependent manner. The fastest pyruvate formation rate was detected at 90°C, where typical metabolic enzymes are not functional. Comparing the three conditions, pyruvate formation rate was lowest in water, increased in the presence of the ocean components and achieved its highest rate in the ferrous iron-rich Archean ocean simulation (Fig 5C). Compared to water at a temperature of 70°C, the pyruvate formation was 49% faster in the presence of the Archean ocean mimetic and ferric iron, and 200% faster in the analogous, ferrous iron containing ocean simulation (Fig 5C). Hence, the ferrous condition plausible for the Archean ocean does not only catalyse interconversion among pentose phosphate sugars, they also favoured specificity and increased reactivity in a non-enzymatic, glycolysis-like, chemical reaction system.

## Discussion

The metabolic network possesses a remarkably similar basic structure in all organisms examined. This indicates that it came into

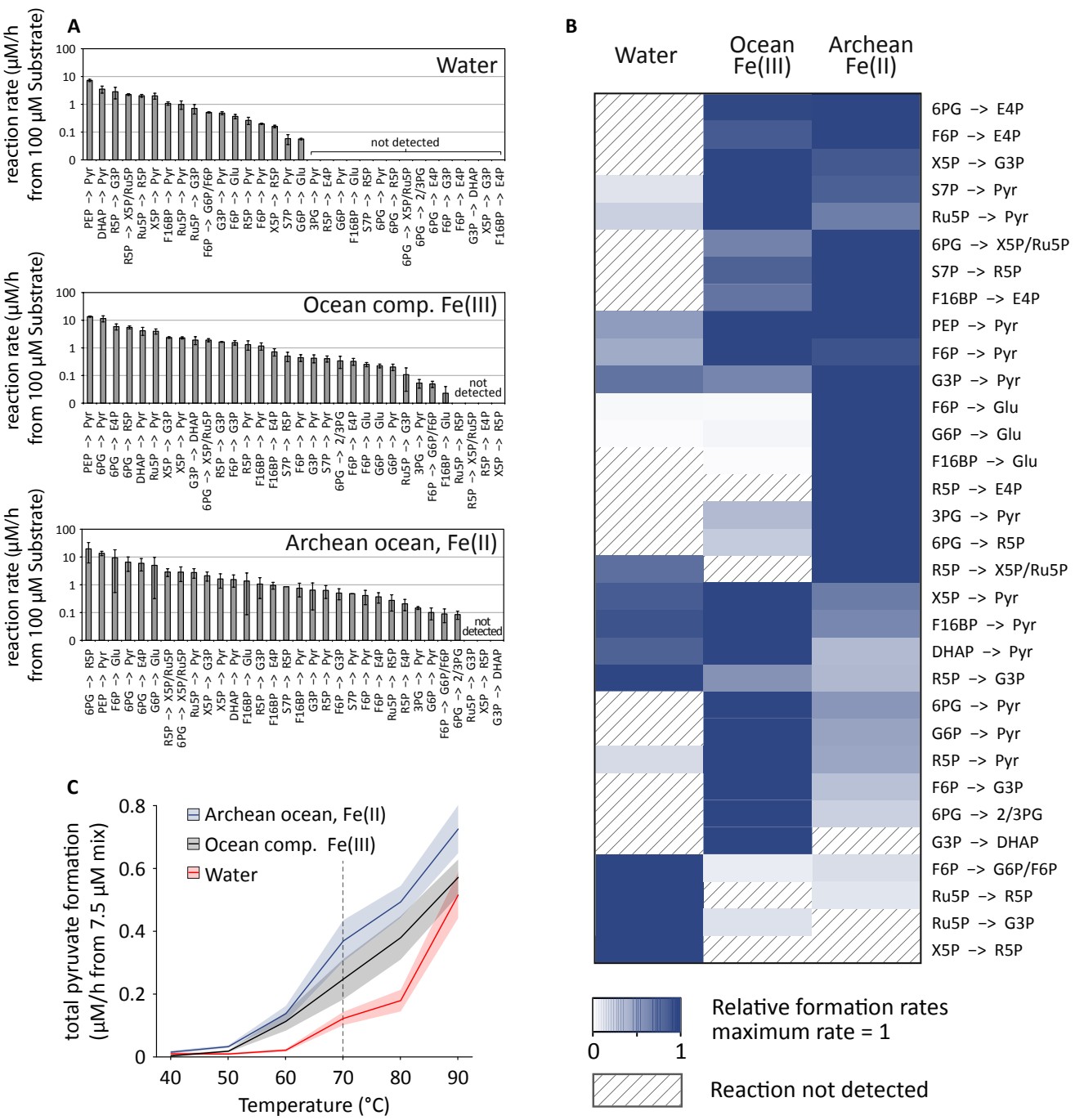

**Figure 5.** **The Archean ocean chemical composition accelerates metabolism-like reactions and the non-enzymatic formation of pyruvate.**

A Reaction rates in water (upper panel), and in the presence of ocean components with ferric iron (middle panel), or ferrous iron (Archean ocean, lower panel). The reaction rates in μM/h were determined by monitoring the formation of metabolites over a 5 h time course, *n* = 3, *y*-axis log scaling. Source data are available in Supplementary Table S5.

B The reaction rates are expressed relative to the condition with the maximum reaction rate (= 1), to allow direct comparison of the rates in water, and in the Archean mimetics. Non-detectable reactions are represented by hatched areas. Source data are available in Supplementary Tables S1 and S5.

C Reactivity within an Archean ocean mimetic accelerates the enzyme-free formation of pyruvate. Pyruvate formation rate in a mixture of pentose phosphate pathway and glycolytic intermediates. Glucose 6-phosphate (G6P), fructose 6-phosphate (F6P), fructose 1,6-bisphosphate (F16BP), dihydroxyacetone phosphate (DHAP), glyceraldehyde 3-phosphate (G3P), 3-phosphoglycerate (3PG), phosphoenolpyruvate (PEP), 6-phosphogluconate (6PG), ribulose 5-phosphate (Ru5P), ribose 5-phosphate (R5P), xylulose 5-phosphate (X5P) and sedoheptulose 7-phosphate (S7P) were each combined at 7.5 μM in: (i) water, (ii) the Archean ocean components with Fe(III) and (iii) the Archean ocean components in the presence of Fe(II). The mixture of the metabolic intermediates was exposed for 5 h at 40–90°C in 10°C steps, and pyruvate formation monitored by LC-SRM. *n* = 3, error areas illustrate ± SEM. Pyruvate formation was not detected below 50°C in water and increased in a temperature-dependent manner, indicative of non-enzymatic reactions. At all temperatures, pyruvate formation was fastest in the Archean ocean reconstruction; at the model temperature of 70°C (highlighted), accelerated by 200%.

being at a very early stage of evolution and that its reaction sequences follow highly optimized routes (Jeong *et al*, 2000; Noor *et al*, 2010). The evolutionary origins of this network structure are, however, still largely unknown (Luisi, 2012).

It is assumed that the pathways that mediate sugar phosphate interconversion, glycolysis, the pentose phosphate pathway, as well as the related Entner-Doudoroff pathway and Calvin cycle are evolutionarily ancient, as they are conserved and fulfil their central metabolic functionality virtually ubiquitously. Known as central, or primary, metabolism, their reaction sequences provide ribose 5-phosphate for the backbone of RNA and DNA, building blocks for the synthesis of co-enzymes, amino acids and lipids and supply the cell with energy in form of ATP and redox equivalents.

Geological records reveal details about the chemical environment under which life spread for the first time. This event has been dated between the earliest Archean eon that followed the late heavy bombardment, likely between 4.1 and 3.5 billion years ago, when the first unequivocal traces of life have been dated (Wacey *et al*, 2011). Geochemical and geological evidence, particularly the large iron isotope fractionations and the lack of sulphur isotope fractionation during pyrite burial, suggests that the oceans of this period were rich in ferrous iron and poor in sulfate. The absolute concentration of iron in these ancient oceans is equivocal with estimates ranging from 20 μM to 5 mM (Canfield, 2000; Saito *et al*, 2003; Rouxel *et al*, 2005; Zerkle, 2005). Knowledge about the concentrations of other bio-essential metals in the Archean oceans during the evolution of life comes from reconstructing the biogeochemical cycles of these elements and taking into account the environmental properties of a world devoid of oxygen (Saito *et al*, 2003; Zerkle, 2005). This has led to the suggestion that copper concentrations in the Archean oceans were negligible and that cobalt and manganese were more abundant than today (but still many orders of magnitude lower than the paleo iron concentrations). The same argument has been applied to zinc; however, its concentration in Proterozoic black shales and Archean iron oxides suggests that its concentration may have remained closer to modern levels over much of the geological time (Scott *et al*, 2012; Robbins *et al*, 2013). Nickel and molybdenum concentrations were likely lower than today, whereas concentrations of cobalt and manganese are assumed to be similar to modern levels (Saito *et al*, 2003; Zerkle, 2005). It has further been suggested that phosphorus is more easily released from sediment in anoxic (or dysoxic) bottom waters, hinting that the Archean oceans may have been rich in phosphate (Van Cappellen & Ingall, 1996).

We used these constraints to reconstruct chemical scenarios plausible for the Archean ocean and tested the effect of this geochemistry on non-enzymatic metabolite interconversion reactions. Importantly, we focussed in these investigations on average Archean ocean ionic compositions. The chemical environments reconstituted are thus plausible for the large majority of the early earth oceans prior to the great oxygenation event and are not restricted to niches on the planet. Triggered by the observation that metabolites that constitute glycolysis and the pentose phosphate pathway spontaneously interconvert in the full absence of enzymes (Fig 1), we tested for an influence and catalytic capacity of the reconstituted oceanic conditions. The ferrous milieu of the reconstructed Archean ocean catalysed a total of 29 metabolism-like interconversion reactions. These included the formation of several glycolytic and pentose phosphate pathway intermediates, such as ribose 5-phosphate and erythrose 4-phosphate.

These reactions were most sensitive to the presence of ferrous iron. This metal was not only abundant in the Archean ocean, it is also known as a catalyst in organic chemical synthesis reactions and further acts as electron donor (Wehrli *et al*, 1989; Rouxel *et al*, 2005; Casey & Guan, 2007). Interestingly, the 29 observed out of 182 theoretical reactions strongly overlap with the enzyme-catalysed reactions of the non-oxidative pentose phosphate pathway and glycolysis. As these reactions dictate a similar network topology indicates that their early structure could have been shaped by the chemical constraints of the Archean ocean.

We are able to rule out enzymatic contamination in our reaction mixes as source of these findings by several means. (i) Interconversion reactions were observed at a temperature that is plausible for the early earth, including ocean environments close to heat sources such as volcanic activity (70–90°C), but which substantially exceeds the heat stability of typical metabolic enzymes; (ii) the interconversion reactions were not observed at temperatures of below 40°C, at which both, common and heat-stable (thermophilic) metabolic enzymes would be functional (see Fig 5B). (iii) Metabolites such as 6-phosphogluconate were stable in the reaction mixtures unless iron or other catalysts were supplemented (Fig 2A); on the other hand, some reactions were not sensitive to the presence of these metals (Figs 1A and 2B); (iv) stoichiometric amounts of enzymatic cofactors [ATP, NAD(H), NADP(H), thiamine diphosphate], required for the function of enzymes, were absent. (v) Specific reaction sequences, such as those representing the oxidative PPP, were absent. (vi) To fully exclude the possibility of contamination with a cofactor-independent multifunctional enzyme that functions at high temperatures but not at 40°C or below, we further filtered the reaction mixtures through Amicon ultracell filters with a mass cut-off of 3.5 kDa, repeated the reactions in the presence of an organic solvent (acetonitrile) and tested different types of reaction tube materials. None of these treatments affected the observed interconversion reactions. Moreover, the reactions were conducted either under conditions where all standard metabolites were treated separately, or in mixtures where they were combined, so that a contamination of a chemical standard with a reactive molecule could not explain the spectra of reactions observed.

One of the difficulties in describing the origin of metabolism is the fact that the metabolic network is largely composed of intermediates that are not characterized by long-time stability, at least when considering geological environments and timescales. As shown here and previously, this in particular applies to sugar phosphate molecules [(Larralde *et al*, 1995), Fig 2]. In addition, large sugar phosphates are not frequently generated in experiments that address scenarios of primordial carbon fixation (Cody, 2000; Fuchs, 2011; Hügler & Sievert, 2011). This difficulty cannot, however, mask the fact that sugar phosphates are constituents of many molecules, such as RNA, DNA, ATP and lipids, which are inevitably connected with the emergence of life. It is the fundamental role of sugar phosphates, and the virtual universality of their few metabolic interconversion sequences, that places their origin to the very early evolutionary stages. Long-term stability seems thus not to be a predictor of whether a molecule adopts a cellular metabolic function. A possible explanation is that molecules stable in a certain environment do not react with their surrounding molecules; inert

molecules were thus unlikely to form reaction systems based on environmental catalysts. This seeming paradox of the universality of sugar phosphates and their low stability in the prebiotic world might be solved by accepting different reaction sequences for carbon fixation and the first forms of metabolism: Carbon fixation could have occurred through non-metabolism-like events, including the so-called formose reaction, which is, in a series of condensation steps, able to convert several formaldehyde molecules into complex carbohydrates structures (Breslow, 1959; Maurer *et al*, 1987), or through alternative/parallel scenarios that include mineral- or photochemically catalysed reactions, as well as microcompartmentalization, that allowed the accumulation of first biomolecules in protocells (Pitsch *et al*, 1995; Cody, 2000; Szostak *et al*, 2001; Bryant *et al*, 2010; Fuchs, 2011; Ritson & Sutherland, 2013). It is only when the first biomolecules had achieved life-compatible concentrations that biologically relevant interconversion sequences, or early forms of metabolic pathways, could become active.

What constrained the first forms of metabolism? Aside the availability of life-supporting concentrations of metabolites, it can be assumed that the presence of catalysts was a major limitation to establish first forms of the metabolic network. Catalysts are required not only to accelerate chemical reactions, but also to achieve specificity in reaction systems—uncatalysed chemical reactions can lead to a large set of unspecific products, while catalysts limit the reaction space by preferring a specific reaction. In modern cells, most metabolic reactions are catalysed by protein-based enzymes. Modern enzymes possess, however, highly specialized and complex structures, and it is unlikely that they evolved before a metabolic system was in place. The major question about the early forms of metabolism is thus the nature about their precursors, the first metabolic catalysts (Lazcano & Miller, 1999; Shapiro, 2000; Anet, 2004; Sousa *et al*, 2013). RNA could be an option; however, central metabolism lacks examples of RNA-catalysed reactions [and synthetic ribozymes that catalyse aldolase reactions require a bivalent metal for its activity (Fusz *et al*, 2005)]. Here, we demonstrate that simple inorganic molecules, frequently found in sediments dated to the Archean period, can catalyse reactions analogous to what is observed in modern pathways. Archean oceanic metals could thus have served as first catalysts in an early form of metabolism. Importantly, as we worked with average concentrations of molecules found on the early planet, the availability for these catalysts was not restricted to niches or extreme conditions—a metabolism that bases on these molecules could thus had occurred throughout the oceanic environment.

How and whether the first enzymes adopted the reaction mechanisms of the metal-catalysed reactions presented here remains an equivocal question. Anecdotal evidence is, however, in favour of such an assumption: Bacteria possess a ribulose 5-phosphate epimerase that uses ferrous iron for its catalytic functionality (Sobota & Imlay, 2011), whereas analogous enzymes in higher eukaryotes are metal independent (Kopp *et al*, 1999). As ferrous iron became limiting after its depletion from oceans during the great oxygenation event, the dependence on ferrous iron, its transport and protective mechanism against oxidation and Fenton reactivity became a strong disadvantage for cells (Askwith & Kaplan, 1998; Adamala & Szostak, 2013). Replacing ribulose 5-phosphate epimerase and other

enzymes by metal-independent enzyme versions would therefore confer selective advantage.

In our simulations, not all reactions of the pentose phosphate and glycolysis pathways were observed. For instance, no direct route of glucose 6-phosphate into 6-phosphogluconate that would correspond to the oxidative pentose phosphate pathway was detected (Fig 1C). Such a reaction may nevertheless occur non-enzymatically under some conditions. We did not, however, optimize the reaction conditions to achieve a maximum number of reactions but we went rather the other way round and conducted a very general, evidence-based ocean reconstruction to test its effects. Therefore, the simulation is by no means mimicking all local environments of the early oceans, neither did we consider mineral surfaces as catalysts that have important capacities as catalysts (Wächtershäuser, 1988). The absence of specific reactions could thus simply indicate their dependence on a different set of ions or constraints that was not included. However, unlike glycolytic reactions and the non-oxidative pentose phosphate pathway, the oxidative pentose phosphate pathway is not universal and is notably absent in many anaerobes and Archaea (Grochowski *et al*, 2005; Nunoura *et al*, 2011). It is thus equally well possible that this part of the pathway came into being with the emergence of enzymes. Therefore, these results could support a hybrid hypothesis to describe the origin of metabolism: a core set of reactions would have be constrained by the environment of the early world organisms, and this network was then extended in terms of both reactivity and efficiency through the evolutionary selection of enzymes until the modern network was in place.

Structural studies of glycolytic enzymes revealed that the main function of the first metabolic network was of an endergonic nature, since gluconeogenesis likely evolutionarily proceeded in the emergence of glycolysis (Say & Fuchs, 2010; Fuchs, 2011). Consequently, the first enzymatic pathways formed could provide the early cells with metabolic intermediates, but supposedly not yet with energy in the form of ATP. Supportingly, the mineral catalysed and non-enzymatic formation of three-carbon sugar phosphates and pyruvate has been described (Pitsch *et al*, 1995; Cody, 2000; Bryant *et al*, 2010), indicating that an early gluconeogenesis could have been possible. Finally, in some thermophilic micro-organisms, a functional 'non-phosphorylating' Entner-Doudoroff pathway has persisted. The pathway that does not yield any ATP from the catabolism of glucose, but provides its advantage by yielding the metabolic intermediates (Reher *et al*, 2010), indicating that a non-ATP-producing glycolysis provides its advantage by providing metabolic intermediates.

The findings reported here have potential implications for the origin of the RNA world. It is reasonable that this biomolecule served as basis of the first genetic organisms, as it is evolutionary ancient, able to encode information, to catalyse chemical reactions and to (self-) replicate (Orgel, 2004). Although there is evidence for the formation of nucleobases under prebiotic plausible conditions, the origin of the sugar phosphate (ribose phosphate) backbone is still debated (Anastasi *et al*, 2008; Powner *et al*, 2009; Hud *et al*, 2013). To our knowledge, all extant organisms completely circumvent the synthesis of free ribose in order to generate ribose 5-phosphate; instead, they use sugar phosphate metabolism and generate this molecule in most cases through the pentose phosphate pathway or the Calvin cycle. The results presented here demonstrate

that sugar phosphate interconversions reactions are prebiotically plausible; thus, the origin of the ribose 5-phosphate through a sugar phosphate interconversion route should be considered. In this context, we have noticed that ribose 5-phosphate was around five times more stable than the other pentose phosphate intermediates (xylulose 5-phosphate and ribulose 5-phosphate) and the formation of ribose 5-phosphate from 6-phosphogluconate was the fastest of all reactions in the presence of Fe(II) (Fig 5A). These properties could have contributed to the central importance of ribose 5-phosphate as backbone of the genetic material.

This study was facilitated by the data-independent nature and sensitivity of the LC-SRM technique. However, only limited conclusions can be drawn about the full spectra of molecules that generate aside the monitored reactions. In the retention times monitored, we did not find evidence for the significant formation of non-glycolytic/pentose phosphate pathway three-, four-, five-, six- and seven-carbon phosphates. This indicates that most carbon equivalents not interconverted into the metabolic intermediates will most likely represent non-phosphorylated sugars and $CO_2$. The full spectra of these reactions need to be clarified in future studies. Moreover, the reactions as presented fully follow the law of thermodynamics, and as consequence, the reaction equilibrium favours intermediates of lower energy. Studies about the thermodynamics in metabolic systems have, however, shown that reaction equilibrium is shifted under conditions that separate a product from the catalyst, when reactions are coupled and when one product is stabilized, that is, by the inclusion in protocell-like vesicles (Wächtershäuser, 1988; Szostak *et al*, 2001; Amend *et al*, 2013). In future studies, it should thus be investigated whether the reactions reported here can be combined with prebiotically plausible mechanisms to stabilize the higher energetic phosphates and thus whether the reactions can be exploited in carbon fixation mechanisms as well.

In summary, we report that plausible Archean ocean chemical compositions serve as catalysts for a series of sugar phosphate interconversions among pentose phosphate pathway and glycolytic metabolites. The reactions connect important metabolic intermediates including ribose 5-phosphate and erythrose 4-phosphate and eventually feed into the metabolite pools of pyruvate and glucose, the stable products of modern glycolysis and gluconeogenesis. These results indicate that the basic architecture of the modern metabolic network could have originated from chemical and physical constraints that existed in the prebiotic earth's ocean. These findings suggest that simple inorganic molecules, abundantly present in the Archean ocean, may have served as catalysts in early forms of metabolism and facilitated sugar phosphate interconversion sequences that resemble glycolysis and the pentose phosphate pathway. These results therefore support the hypothesis that the topology of extant metabolic network could have originated from the structure of a primitive, metabolism-like, prebiotic chemical interconversion network.

# Materials and Methods

## Metabolite standards

Metabolite standards were purchased at highest available quality from Sigma-Aldrich; fructose 1,6-bisphosphate (F16BP, order number F6803), fructose 6-phosphate (F6P, F3627), dihydroxyacetone phosphate (DHAP, D7137), glucose 6-phosphate (G6P, G7879), 3-phosphoglyceric acid (3PG, P8877), 6-phosphogluconic acid (6PG, P6888), ribose 5-phosphate (R5P, R7750), xylulose 5-phosphate (X5P, X0754), ribulose 5-phosphate (Ru5P, 83899), phosphoenolpyruvate (PEP, P7127), glucose (Glu, 16325), pyruvate (Pyr, P2256), sedoheptulose 7-phosphate (S7P, 78832) and glyceraldehyde 3-phosphate (G3P, G5376). Information about the standards purity is provided in the Supplementary Table S6. Please note that the highest purity available for sedoheptulose 7-phosphate was 90%; reaction rates determined for this metabolite may thus be influenced by impurities. Water (TOC < 10 ppb, Resistivity > 18.2 MΩ*cm, Greyhound Chemicals, BIO-23214102) and acetonitrile (Greyhound Chemicals) was used exclusively in ULC-MS grade. Octylammonium acetate was prepared from octylamine (Sigma-Aldrich) and acetic acid as previously described (Wamelink *et al*, 2005). The metabolite standards were diluted to 100 μM concentrations or combined at 7.5 μM each as indicated, sealed in HPLC vials (5182–0717 & 5182–0716, Agilent Technologies) and heated to 70°C in a water bath. Sampling was conducted in a time series after 0, 10, 30, 60, 120 and 300 min by rapid cooling of the samples. The influence of modern sea salts (23.9 g/l NaCl, 0.677 KCl, 0.196 $NaHCO_3$, 0.026 $H_3BO_4$, 0.003 NaF, 53 mM $MgCl_2$, 10.3 mM $CaCl_2$) or plausible Hadean sea salts (100 mM $MgCl_2$, 50 mM $CaCl_2$, saturated $Na_2SiO_3$) on sugar phosphate interconversion was tested in combination with four different concentrations of metal ions and phosphate (0–200 μM $FeCl_2$, 0–10 nM $MoO_4$, 0–400 nM $NiCl_2$, 10 nM $CoCl_2$, 0–100 μM $H_3PO_4$). From these tests, we selected a buffer containing 200 μM $FeCl_2$, 10 nM $CoCl_2$, 400 nM $NiCl_2$, 10 nM $MoO_4$ and 100 μM $H_3PO_4$ for the Archean ocean simulations. The oxidation state of the solution was confirmed prior to the experiment by formation of an iron–thiocyanate complex, and comparison to $FeCl_2$ and $FeCl_3$ was used to monitor the depletion of detectable ferrous iron in the oxic solution. The anoxic experiments were conducted in an anoxic chamber (Coy Lab Products, USA). Buffer and sugar phosphate mixtures were freshly prepared and then transferred immediately into the anoxic chamber. Oxygen was removed from samples with three repeated vacuum/$N_2$, $H_2$ gas cycles.

## LC-MS/MS measurements

Quantification of sugar phosphates was conducted with a method modified from our earlier procedures (Wamelink *et al*, 2005, 2009; Ralser *et al*, 2009), using an online coupled LC system (Agilent 1290) and triple quadrupole mass spectrometer (Agilent 6460) operating in SRM mode. The 182 theoretically possible reactions were monitored in all samples using Mass Hunter Workstation (Agilent) by separating the samples on a $C_8$ column [Zorbax SB-$C_8$ Rapid Resolution HD, 2.1 × 100 mm, 1.8 μm (Agilent); column temperature: 20°C]. Buffer A and B consisted of 10% and 50% acetonitrile, respectively, and both contain 750 mg/l octylammonium acetate. Separation was achieved by isocratic flow (flow rate 0.6 ml/min) at 5% B for 3.5 min, then ramping to 70% B within 2.5 min, followed by washing with 80% B for 0.5 min and re-equilibration at 5% B for 0.5 min, resulting in a total cycle time of 7.5 min. Metabolites were identified by matching retention time and fragmentation pattern with the commercially obtained standards as indicated in Supplementary Table S4. Nebulizer and drying gas parameters as well as

ionization and fragmentation energies were optimized for the individual metabolites. Ion source settings are listed in Supplementary Table S3, and SRM transition and instrument settings are given in Supplementary Table S4.

Raw MS/MS data were analysed with Masshunter Workstation (Agilent). The automated peak integration was reviewed manually for all metabolite measurements. Peak areas were, when needed, corrected by referring to repeatedly measured quality control samples. As pentose sugar phosphates trigger a low signal within the G3P transition 169.0→97.0, G3P concentrations were corrected by subtracting the maximum observed contribution of these signals. Each of the replicates contained time-series data for the 182 out of the 210 theoretically possible reactions, which were individually analysed with R (R Core Team, http://www.R-project.org). The following fitting models address the different observed reaction modes of product accumulation (which were dependent on substrate, product, co-product and intermediate stabilities and their respective reaction orders): (i) least squares linear model for continuously accumulating metabolites; (ii) nonlinear growth model with the R module SSgompertz for metabolites reaching a steady state; (iii) maximum initial growth rate within the linear range using the least squares model for products with dynamic formation/degradation behaviour; (iv) exponential decay model for first-order-like degradation of educts. The best fitting model was selected based on $R^2$ values and was verified by graphical examination of the fitted curves. Qualitative and quantitative description of the observed reactions was extracted from the respective fitting model (Supplementary Tables S1 and S2). All thermal interconversion experiments were conducted in triplicates at a minimum, and time-dependent accumulation of interconversion products detected in at least three independent experiments was considered as significant. In total, 1200 samples were processed, and as additional quality control, 50% of the samples were measured in technical replicates to exclude variation in instrument. The concentrations obtained in the quantification experiments are provided as Supplementary Table S5.

**Supplementary information** for this article is available online: http://msb.embopress.org

## Acknowledgements

We thank our laboratory members, Prof. Steve Oliver (Univ. of Cambridge) and Duncan Odom (Cancer Research UK Cambridge Institute) for invaluable critical discussions and to George Averill (Univ. of Cambridge) for help in proofreading of the manuscript. We acknowledge funding from the Isaac Newton Trust (RG 68998 to MR), the Wellcome Trust (RG 093735/Z/10/Z to MR), the Royal Society (RG60279 to AVT) and the ERC (Starting grant 260809 to MR and Starting Grant 307582 to AVT) for funding. MAK is supported by an Erwin Schroedinger postdoctoral fellowship (Austria) (FWF Austria, J 3341). MR is a Wellcome Trust Research Career Development and Wellcome-Beit Prize fellow.

## Author contributions

MAK conducted the experiments, data analysis and illustration, and MAK, AVT and MR designed the study and experiments. MR wrote the first draft of the paper, and all authors contributed to writing and finalizing the manuscript.

## Conflict of interest

The authors declare that they have no conflict of interest.

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
