## [Review Process File · Molecular Systems Biology]

Spontaneous assembly of a glycolysis-like reaction network in the plausible primeval ocean

Markus A. Keller, Alexandra V. Turchyn and Markus Ralser

Corresponding author: Markus Ralser, University of Cambridge

Review timeline:

Submission date:	24 February 2014
Editorial Decision:	27 February 2014
Revision received:	06 March 2014
Editorial Decision:	10 March 2014
Revision received:	11 March 2014
Accepted:	18 March 2014

Editor: Thomas Lemberger

Transaction Report:

1st Editorial Decision

27 February 2014

Thank you again for submitting your work to Molecular Systems Biology. We have now studied your transferred manuscript and the accompanying reviews. While reviewer #2 is clearly positive, reviewer #1 is much less supportive. While the opinions about the interpretation of the data are split, we feel that the results presented are thought-provoking and might be of interest to a broad community of life scientists, including systems biologists and synthetic biologists. Given that the data are not challenged by the reviewers and that most of the criticisms refer to the broader interpretation of the data, we feel that we can consider a revision of the present work, provided the manuscript is revised to address the reviewers' comment and in particular the following major points:

- In view of the comments provided by the reviewers, we would ask you to present the data in the Results section in a rigorous and as neutral as possible manner. Please move any interpretation and speculations to the Discussion section. Whenever possible, the results should be described with specific rather than general statements (eg "reaction rates were increased 2 fold" instead of "reaction rates were markedly increased"). Please avoid the use of terms like 'dramatically', 'strikingly', 'remarkably' in the Results section.
- It should be made much clearer how many metabolites (and which one) were actually measured by SRM in the different experiments reported in this study. This is important to provide context to statements about the number of reactions and metabolites observed under specific conditions. This information can be added to the main text and the figure legends as well.
- A better visualization could perhaps be provided for some time-course experiments to provide a sense of the sequence of events occurring during the experiment.

- Please indicate in figure legends how data are plotted (as average?) and how is the variability of the data estimated. This particularly relevant for Fig 4b. Statistical tests of significance should be provided when appropriate.

- The manuscript refers to the 'spontaneous assembly of a glycolysis-like reaction network' and highlights the 'resemblance' of the set of abiotic reactions with the corresponding in vivo counterparts. Since your experimental design is targeted in nature, the 'topology' of the observed set of reactions is partly determined by which set of substrates you decided to test in the first place and which subset of products were measured by targeted methods. While it seems justified and relevant to highlight the fraction of glycolytic-like reactions observed and the significant recovery rate, and hence the resulting overlap with the in vivo counterpart, reference to the 'spontaneous assembly' of an emerging network 'topology' could potentially mislead some readers. We think that the data speak for themselves and we would thus ask you to go over the entire text, title, abstract and figure legends, to take this point into consideration and use rigorous wording. We provide also suggestions in the attached Word file.

- It would be important to include a 'caveat' section in the Discussion to explicitly list immediate issues that remain open -- for example:

--- the uncertainty about the availability of precursors in 'sufficient amounts in prebiotic environments';

--- the nature of the newly formed metabolites that were not measured here (40-50% of the recovered carbon) and how complex is this mixture;

---what are the implications of the 'missing link' from G6P to 6PG;

---is there CO₂ production?;

---what about the reversibility/irreversibility of the reactions involving C-C bond formation/splitting?

- Citation to some of the early work by Waecktershaeuser (PNAS 1990 or Microbiol Rev 1988) on the evolution of the first metabolic cycles could be added.

- The discussion is very long and could be shortened somewhat to make it more compact and focused on the most important points.

- We would also ask you to provide the SRM data as dataset files (or to submit them to an appropriate public database; would MetaboLight be appropriate?).

Please note that we also attach the Word file of the manuscript with suggestions for edits and amendments. We would be grateful if you could go through these when revising the text. We have also suggested to include source data files for specific panels to allow readers to visualize the data in alternate manner and re-analyze them (see also our guidelines: <http://msb.embopress.org/authorguide#a3.4.3>).

Please resubmit your revised manuscript online, with a covering letter listing amendments and responses to each point raised by the referees.

TRANSFERRED REVIEW #1

In the present work, the authors investigate the enzyme-free conversion of glycolysis and pentose phosphate pathway intermediates. Using targeted mass-spectrometry, the authors show that heat treatment triggers a number of conversion reactions in absence of enzymes, including isomerization of hexose- and pentose-phosphates, and in particular the formation of pyruvate. Moreover, the authors show that addition of plausible archean ocean salts increases the stability of the majority of metabolites, as well as the specificity of the conversion reactions judging from the higher formation rates.

This work constitutes an interesting experimental approach to study the constraints and properties of non-enzymatic metabolism and is therefore surely interesting for a broader audience. However, there are several comments which should be addressed by the authors before I can recommend this work for publication.

1. The achievements are marginal. The authors essentially show that sugar phosphate intermediates from glycolysis and pentose phosphate pathway are degraded mostly to smaller glycolytic intermediates upon heat exposure (and in absence of enzymes), which is probably not entirely surprising. In fact, the fast degradation of sugar phosphates has been observed long ago, albeit without reporting the degradation products (Larralde et al., 1995; PMID 7667262). There is in fact no evidence for aldolase/ketolase-like condensation reactions to produce C-C bonds, or phosphorylation. These are key steps of glycolysis/gluconeogenesis/pentose phosphate shunt. The link to a potential prebiotic metabolism is weaker than the authors claim.

2. Along the same lines, the study is flawed by the fact that there has been no report on the feasibility of non-enzymatic formation of 6-phosphogluconate or pentose-5-phosphates under prebiotic conditions. However, in the present work these metabolites are the exclusive sources for ribose-5-phosphate, which the authors highlight in their discussion due to its role in RNA. Thus, the results presented here are only relevant for prebiotic metabolism under the assumption that these "modern" sugar phosphates were present continuously in sufficient amounts in prebiotic environments, which is far from certain.

3. What are 5 hours in the span of evolution? I challenge the view that absence of detectable glycolytic products in water proves anything. The same glycolytic-like network topology could also arise in a water milieu at a much lower pace. The authors argue that iron supports a more specific conversion between glycolytic intermediates, but the effects are weak (20 vs 25%, according to the numbers reported on page 11 referring to figure 4a).

4. The authors use targeted metabolomics and restrict themselves to the quantification of only those metabolites which they tested for their reactivity. However, the measured compounds seem to account for only 20-25% of the reaction products (according to the numbers reported on page 11). Hence, the majority of alternative products was not detected. Are these unknown products a wild mix of different species or are they dominated by specific compounds? By restricting the analysis to targeted metabolomics, the authors seem to miss a good opportunity to discover important intermediates which may not be part of modern metabolism and therefore constitute non-enzymatic reactions which form a prebiotic "metabolism as we don't know it", to paraphrase Günter Wächtershäuser (PMID 10979855). In this view, the increased stability of sugar-phosphate in iron II is not necessarily a positive aspect.

5. Many of the author's claims, in particular regarding the role of abiotic catalysts, rely on the quantification of conversion rates. Although the authors do provide the calculated conversion rates in the supplement, showing the corresponding time course data too (at least for the data shown in figure 4c) would allow to better appreciate the dynamics of metabolite conversions (e.g. to assess whether there is any temporal order in the conversion reactions).

6. The sugar-phosphates used are delivered in different salt forms (3Na^+ , 2Na^+ , Na^+ , Li^+ , K^+ , diethylacetate- Ba^+). Given the amounts used (100 μM), the salts included in the substrate are likely to differentially change the composition of the Archean buffer. Moreover, some substrates have only 90% purity. What countermeasures and tests were implemented to exclude artifacts?

7. The discussion is too long and could benefit from better structuring. Moreover, the discussion of the implications on "origin-of-life" theories should be condensed.

Additional comments:

1. I was confused by figure 1: why is it split in two partly redundant panels?

2. There are a number of metabolites whose non-enzymatic formation has been reported, e.g. ribose-2,4-bisphosphate, hexose-2,4,6-triphosphate (Mueller et al., 1990; *Helvetica Chimica Acta* 73:1410-1468) or pyrimidine ribonucleotides (Powner et al., 2009; PMID 19444213). In particular the former

constitute non-typical metabolites which are in our current understanding not part of modern metabolic systems. Is it possible to test any of these compounds for their degradation products to check whether they are eventually converted into glycolysis or PPP metabolites?

3. The authors conclude that some metal ions, in particular Fe(II), act as catalysts in the conversion reactions (thereby increasing their specificity). Can the authors exclude that Fe(II) is not a catalyst, but is rather being consumed during the conversion reactions in an unknown mechanism (since Fe(II) is present at higher concentrations (200 μ M vs. 7.5/100 μ M) than the metabolites)? For example, do the conversion rates depend on the concentration of Fe(II), which would indeed suggest that it acts as a catalyst?

4. Page 20 lines 9-16: the authors discuss the apparent paradox that thermophile organisms can thrive at temperatures which exceed their metabolite's stability. It is not clear to me why this is necessarily a paradox: the turnover rates of metabolites in glycolysis are in the order of seconds for model organisms such as E.coli. Even if one assumes turnover rates which are 1-2 orders of magnitude lower, these would still be much higher than the degradation rates reported here, suggesting that metabolite degradation is negligible compared to enzymatic conversion, at least in actively metabolizing cells. The authors should make this aspect clear in the discussion.

5. Some of the conversion rates are in the nano-molar range - how did the authors decide whether a particular metabolite is formed at such slow rates vs not formed at all? Did they use a metabolite-specific detection limit, or a general concentration cut-off?

TRANSFERRED REVIEW #2

This article presents some surprising and unexpected results about the stability and spontaneous chemical conversions of intermediary metabolites that has relevance to the evolution of metabolism. Though one might worry about the presence of traces of enzymes remaining in the metabolite samples, the authors have excluded this as a plausible explanation of the results.

Major point

1. There is a weakness in the argument on p. 19 that these results could account for the formation of ribose 5P, since net formation of pentose sugars has only been shown to occur from 6-phosphogluconate. Now there is no interconversion shown that generates 6-phosphogluconate. Indeed, the authors themselves argue that the oxidative pentose phosphate pathway is not universal in archaea and could have arisen later. Organisms that lack the oxidative pentose pathway can form ribose 5P from hexoses and trioses, but the necessary conversions are not present in their demonstrated reaction set. This inconsistency in the argument should certainly be removed, but in my view, they have not provided any new support for this weak point of the RNA world hypothesis.

Minor points

2. Sentence at lines 17-19, p. 8. I was unclear what was implied by this sentence, since it seems to contradict the following sentence. Do they mean they saw no significant differences unless they included the metal ions listed? If so, that would be a shorter and clearer statement.

3. The text needs significant sub-editing to remove disagreements between subjects and verbs, changes of tense within sentences and typographic errors.

4. The reference for Fuchs (2011) is incomplete.

Thank you very much for giving us the opportunity to submit a revised version of our article "Non-enzymatic glycolysis and pentose phosphate pathway-like reactions in the plausible Archean ocean"

We thank the Reviewers and the Editorial team for providing many productive suggestions to improve the paper. We detail below how we incorporated the suggestions, in particular we focused in improving the description of the used methods and standards. We hope that our manuscript fulfills now the requested criteria, and would be delighted to see it published with MSB.

Response to Editor and Reviewers:

- In view of the comments provided by the reviewers, we would ask you to present the data in the Results section in a rigorous and as neutral as possible manner. Please move any interpretation and speculations to the Discussion section. Whenever possible, the results should be described with specific rather than general statements (eg "reaction rates were increased 2 fold" instead of "reaction rates were markedly increased"). Please avoid the use of terms like 'dramatically', 'strikingly', 'remarkably' in the Results section.

The text has been revised accordingly. In particular we focused on a neutral presentation in abstract and Results section. In order to address questions of the Reviewers and Editors, we however included a new speculative paragraph about the metabolite stability in the Archean the into the Discussion section.

- It should be made much clearer how many metabolites (and which one) were actually measured by SRM in the different experiments reported in this study. This is important to provide context to statements about the number of reactions and metabolites observed under specific conditions. This information can be added to the main text and the figure legends as well.

We apologize that the used methods were not as clearly described as required. In every SRM measurement, we targeted all intermediates of glycolysis and the PPP. The method was able to quantify 15 metabolites, that represents full coverage of the pathways except one intermediate of glycolysis (1,3- bisphosphoglycerate) and one pentose phosphate pathway intermediate (6-phospho-glucono-delta-lactone) which were not detectable with our setup. The respective paragraph was extended in the manuscript, and in several occasions we now refer to the number of metabolites quantified and the total number of possible reactions monitored, which is 182 for every experiment.

- A better visualization could perhaps be provided for some time-course experiments to provide a sense of the sequence of events occurring during the experiment.

A classical time course experiment is illustrated in Figure 3 for mentoring the concentration of G3P over 5 hours in water and the two Archean reconstructions. This diagram is representative for other sugar phosphates, therefore for them we give the rates only (Fig 3b). For the formation rates that have been obtained in the time course experiments, we now include novel barcharts in Figure 5a, that illustrate and rank the reaction rates for all interconversions observed. Furthermore the reactions as expected follow different rate laws. This is listed for every reaction separately in Suppl Table 1. We hope this fulfills the requirements of the Editor, we are however happy to provide other diagrams if specified.

- Please indicate in figure legends how data are plotted (as average?) and how is the variability of the data estimated. This particularly relevant for Fig 4b. Statistical tests of significance should be provided when appropriate.

We revised the Figure legends and description accordingly. Statistical methods and measures used to determine the reaction rates are given in Supplementary Table 1 and Table 2. All experiments have been performed in a minimum of a triplicate; error bars have been included in all figures and represent SD or SEM as indicated.

We considered to add an additional statistical test for significant differences among the oceanic conditions to Fig 5c, however it is not essential in this figure, and to avoid large confusions for the following reason we would opt out against: Reaction rates were determined by fitting the formation curves to a) least squares linear model for continuously accumulating metabolites; b) non-linear growth model with the R module SSgompertz for metabolites reaching a steady state; c) Maximum initial growth rate within the linear range using a least squares model for products with dynamic formation/degradation behaviour; d) Exponential decay model for first-order like degradation of educts. Due to the different connectivity of a metabolite in the different simulations (water, Archean ocean Fe(II), Archean ocean Fe(III)), the formation or conversion of the same metabolite can follow

a different rate law in the different condition. Consequently, not an uniform statistical test can be applied. What we did as alternative: we did include the statistical variance in a new bar chart included in Figure 5a as error bars \pm SEM. The error bars now illustrate how exact the determination of the rate law was. All other statistical information, and how the rates are calculated, is given in Suppl table 1, raw data is provided in Suppl table 5.

- The manuscript refers to the 'spontaneous assembly of a glycolysis-like reaction network' and highlights the 'resemblance' of the set of abiotic reactions with the corresponding in vivo counterparts. Since your experimental design is targeted in nature, the 'topology' of the observed set of reactions is partly determined by which set of substrates you decided to test in the first place and which subset of products were measured by targeted methods. While it seems justified and relevant to highlight the fraction of glycolytic-like reactions observed and the significant recovery rate, and hence the resulting overlap with the in vivo counterpart, reference to the 'spontaneous assembly' of an emerging network 'topology' could potentially mislead some readers. We think that the data speak for themselves and we would thus ask you to go over the entire text, title, abstract and figure legends, to take this point into consideration and use rigorous wording. We provide also suggestions in the attached Word file.

We followed the Editors suggestion and revised our text accordingly. We propose the title “Non-enzymatic glycolysis and pentose phosphate pathway-like reactions in the plausible Archean ocean”

- It would be important to include a 'caveat' section in the Discussion to explicitly list immediate issues that remain open -- for example:

--- the uncertainty about the availability of precursors in 'sufficient amounts in prebiotic environments';

We fully agree that the question how life supporting concentrations of the first molecules of life on earth can have reached, primordial carbon fixation, is among the major unsolved issues regarding the origin of life. This answer is not given in the present paper either. This is now explicitly stated in the discussion and respective literature cited. In addition we include a new paragraph to provide an attempt for a hypothesis about the unsolved question (not only in this work, but about the origin of life in general) that cellular metabolism and cellular macromolecules (i.e. RNA) is constituted of metabolites that are not stable enough to persist over long geological timeframes.

--- the nature of the newly formed metabolites that were not measured here (40-50% of the recovered carbon) and how complex is this mixture;

---is there CO₂ production?;

This issue is now explicitly discussed. The SRM method has the advantage over discovery metabolomics that it is more sensitive, absolute quantitative for all studied metabolites, an almost full coverage of intermediates of a pathway is achieved. Due to the data independent nature of the SRM assays, it is possible to make conclusions not only about the presence, but also about the absence of a molecule (given the detection limits of the platform). For the latter two reasons the present study was **only made possible** with a targeted SRM method, and could not have been conducted with a discovery setup. Please also consider the required throughput of > 2000 samples that had to be measured. Certainly however, the SRM method is not a discovery method suitable to dissect the structure of additional metabolites formed. We clearly state in the manuscript that glycolytic and PPP intermediates are not exclusively formed. From matching retention times, we can however exclude that dominant quantities of non glycolytic and PPP intermediate pentose and hexose sugar phosphates are formed aside the glycolytic/PPP intermediates. Similarly we did not observe additional three-carbon phosphates in the broad retention time windows monitored. It is thus to be assumed that the majority of the carbon which does not interconvert into glycolytic PPP intermediates yields the formation of CO₂, the energetically favored product of carbon decomposition, and de-phosphorylated sugars. This information has been included in the revised version of the text.

---what are the implications of the 'missing link' from G6P to 6PG;

This route is in modern cells known as the oxidative PPP. The oxidative PPP is not only absent in our simulation, but also in many microbial species and many Archaeae. The fact that the oxidative PPP is not universal indicates that it came into being at a later stage in evolution. This is discussed in the article. Reviewer #1 who brought up this point that only the oxidative PPP provides ribose 5-phosphate but overlooked the fact that in many organisms it is the non-oxidative PPP that dominates

in this function, in some the oxidative part is indeed completely absent (Clasquin et al., 2011)(Nunoura et al., 2011).

---what about the reversibility/irreversibility of the reactions involving C-C bond formation/splitting?

Our work does not answer the question how prebiotic carbon fixation occurred. However, the metals catalyze similar interconversions as enzymes in glycolysis and the PPP do. There was extensive previous work on metabolic reactions, demonstrating that that they are equilibrium reactions (Fraenkel, 1986). Although in (the minority) of the cases the thermodynamically equilibrium is strongly at the product side, a back-reaction is fully within thermodynamic principles. A situation where the thermodynamically less favored metabolite would be metabolized further, or separated from the catalyst, would thus facilitate its accumulation (Szostak, Bartel, & Luisi, 2001). We include now into discussion that these situations could lead to carbon fixation.

- Citation to some of the early work by Waechtershaeuser (PNAS 1990 or Microbiol Rev 1988) on the evolution of the first metabolic cycles could be added.

We thank for the suggestion, these References have been included.

- The discussion is very long and could be shortened somewhat to make it more compact and focused on the most important points.

The discussion has been thoroughly revised and non-essential aspects have been removed. As additional aspects had however to be discussed due to questions of the Review process and migration of interpretive parts from the Results section, the discussion is however not shorter in total word count. A major point of discussion is now how the origin of the metabolic network is compatible with the low stability of its metabolic intermediates, also technological aspects are more intensively discussed.

- We would also ask you to provide the SRM data as dataset files (or to submit them to an appropriate public database; would MetaboLight be appropriate?).

We include the raw data as a supplementary dataset (Supplementary Table 5) into the revised version.

Specific comments to reviewers:

Response to TRANSFERRED REVIEW #1

...the fast degradation of sugar phosphates has been observed long ago, albeit without reporting the degradation products (Larralde et al., 1995; PMID 7667262).

We included this Reference to the manuscript. However, without reporting the products formed, the Reference provided does not add significant value to the conclusions of this manuscript.

There is in fact no evidence for aldolase/ketolase-like condensation reactions to produce C-C bonds, or phosphorylation. This are key step of glycolysis/gluconeogenesis/pentose phosphate shunt.

Please see answer to the Editor: Research about the biochemistry of metabolism has shown that all reactions of glycolysis and the PPP are equilibrium reactions, despite in some cases the one of the products is energetically strongly favored (Amend, LaRowe, McCollom, & Shock, 2013; Fraenkel, 1986). Enzymes act as catalysts, they can not make reactions happen that are thermodynamically not possible. What is true for enzymes also applies to the small molecule catalysts we identified: they can catalyze the reactions, but they do not influence the physical properties of the reaction substrates or products, nor their thermodynamic equilibrium. So in the end it does not matter for the physical property of the substrate or the product whether the reaction is catalyzed by an enzyme or a chemical catalyst. However, there are systems in place which shift the reaction equilibrium; i.e. an attractor can remove one product from the catalyst (Le Chatelier), leading to accumulation of the product despite this product is not favoured in the equilibrium. Here we focused on identifying the catalysts, but we agree that future work should concentrate on situations that influence thermodynamic equilibria, i.e. to allow carbon fixation (Amend et al., 2013; Szostak et al., 2001). The discussion has been extended to clarify this point.

2. ...there has been no report on the feasibility of non-enzymatic formation of 6-phosphogluconate or pentose-5-phosphates under prebiotic conditions. However, in the present work these metabolites are the exclusive sources for ribose-5-phosphate, which the authors highlight in their discussion due to its role in RNA. Thus, the results presented here are only relevant for prebiotic metabolism under the assumption that these "modern" sugar phosphates were present continuously in sufficient amounts in prebiotic environments, which is far from certain.

The question how prebiotic carbon fixation could have occurred is a comprehensive one that puzzles a whole field of Research since decades. Especially regarding sugar phosphates major questions are still unproven, i.e. its not even known how or if the RNA backbone molecule ribose 5-phosphate could have constituted (Braakman & Smith, 2012; Fuchs, 2011). We agree that it is not yet clear how first sugar phosphates have been formed, and that this question is also not answered in our paper. As anecdotal evidence to illustrate how difficult the question is, the field working on the RNA world tries since three decades to come up with a plausible prebiotic RNA molecule, or a pentose phosphate, and was so far not convincingly successful.

Here we deal with a certainly related, but further downstream question. Assuming the metabolites exist, how did then the metabolic network start? Despite we consider the origin of carbon fixation out of scope for this paper, we fully agree with its importance, and as detailed in the discussion that our finding indeed may open a new perspective also for this: Sugar phosphate interconversion reactions could have helped to interconvert different sugar phosphates. By showing that primordial plausible catalysts for glycolysis and PPP like reactions exists provides a new possibility how the structure of the two pathways could have formed.

...The same glycolytic-like network topology could also arise in a water milieu at a much lower pace.

We have no experimental evidence for this statement. But, the definition of a catalyst is certainly that it accelerates the kinetics of a reaction out of many possible, it does not make thermodynamically impossible reactions possible. However the fact that a reaction is possible does not predict whether it happens under the given conditions, i.e in water (Amend et al., 2013). I hope the Reviewer allows an illustrative example? A student may forget to put the polymerase (the catalyst) into the PCR reaction. Despite all components are there, and the reaction is possible, the DNA template will not get amplified, irrespective of the time allowed. Thus, in the end the polymerase, the catalyst, is important for the reaction.

The conceptual advantage of this study is to identify the oceanic metals as catalysts for the glycolytic reactions. Thus this study implies that these reactions were possible in the Archen ocean, even though modern enzymes were presumably not yet in place.

The authors argue that iron supports a more specific conversion between glycolytic intermediates, but the effects are weak (20 vs 25%, according to the numbers reported on page 11 referring to figure 4a).

The authors use targeted metabolomics and restrict themselves to the quantification of only those metabolites which they tested for their reactivity. However, the measured compounds seem to account for only 20-25% of the reaction products (according to the numbers reported on page 11). Hence, the majority of alternative products was not detected. Are these unknown products a wild mix of different species or are they dominated by specific compounds? By restricting the analysis to targeted metabolomics, the authors seem to miss a good opportunity to discover important intermediates which may not be part of modern metabolism and therefore constitute non-enzymatic reactions which form a prebiotic "metabolism as we don't know it," to paraphrase Günter Wächtershäuser (PMID 10979855). In this view, the increased stability of sugar-phosphate in iron II is not necessarily a positive aspect.

We respectfully disagree. A yield of 25% for a (not-for-yield optimized (!!)) organic chemical reaction system that involves 29 reaction steps is by all terms of measure efficient. Enzyme catalyzed glycolysis is in fact not much more efficient, at least when conducted *in vitro* (Itoh et al., 2004). The redox state of iron alone contributed to ¼ of this value.

We would like to refer to the comments to the Editor why we choose a targeted method. The most important reasons were *i)* that our study required absolute quantity for all metabolites determined in order to be able to calculate reaction rates. Moreover, *ii)* we also needed not only to make conclusions about the presence, but also the absence of a molecule, which required a data independent acquisition approach. Third, *iii)* it was required to have a sensitive analytics that quantifies all metabolites that constitute the pathway at the same time. Methods like NMR are not

targeted, but considerably less sensitive, so it is not possible to get the full coverage of glycolysis and the PPP metabolites in every sample. *iii*) moreover the Reviewer needs to consider that this study required the analysis of more than 2000 samples, not achievable by the throughput of most methods except targeted LC-MS/MS, and certainly not by NMR. Thus, the SRM technology was the only choice to conduct this study.

We agree that LC-SRM does however not allow many conclusions about other metabolites formed. Some however are possible, most importantly that no dominant other 3-6C sugar phosphate metabolites were detected in the retention times monitored. We thus have founded reason to speculate that most other metabolites formed were CO₂ and non phosphorylated sugars. This information has been included in the discussion.

5. Many of the author's claims, in particular regarding the role of abiotic catalysts, rely on the quantification of conversion rates. Although the authors do provide the calculated conversion rates in the supplement, showing the corresponding time course data too (at least for the data shown in figure 4c) would allow to better appreciate the dynamics of metabolite conversions (e.g. to assess whether there is any temporal order in the conversion reactions).

We included a barchart illustrating all reactions in Figure 5a. We agree the reactions follow different functions, so there is time dependence. The models that describe the reactions best, and thus describe which time course the different reactions follow, are listed in Suppl table 1.

6. The sugar-phosphates used are delivered in different salt forms (3Na⁺, 2Na⁺, Na⁺, Li⁺, K⁺, diethylacetale-Ba⁺). Given the amounts used (100 microM), the salts included in the substrate are likely to differentially change the composition of the Archean buffer. Moreover, some substrates have only 90% purity. What countermeasures and tests were implemented to exclude artifacts?

We tested a large series of salts/ions on the reactivity of sugar phosphates (See Results section). We found reactivity only for Fe, Co, Ni, and phosphate, whereas iron dominated the reaction system. These ions were not contained in the metabolites obtained by the supplier. All metabolite were purchased at the highest purity available (we provide the order Reference for standards. To make this point more clear, we now include a Suppl. Table 6 and list all standard purities and their order Reference). We regret that one of the metabolites (sedoheptulose 7-phosphate) was only available at 90% purity. We added a caveat that the reaction rates determined for sedoheptulose 7-phosphate could be confounded as this compound is not available in full purity. The reason for this is however based not on flawlessness of Sigma-Aldrich or the synthesis company: the complex chemistry of a 7C sugar phosphate makes it very difficult to achieve a higher stereochemical purity in its synthesis. We had to decide whether include sedoheptulose 7-phosphate or not for this reason. But as it is a crucial intermediate and we have seen no evidence that it behaves different to other PPP intermediates we concluded that the advantage of picturing all reactions of the PPP outweighs the slight caveat that the reaction rates concerning this particular metabolite can be influenced by the optical purity of the standard.

7. The discussion is too long and could benefit from better structuring. Moreover, the discussion of the implications on "origin-of-life" theories should be condensed.

We revised the discussion accordingly.

Additional comments:

1. I was confused by figure 1: why is it split in two partly redundant panels?

This was requested in a previous review process to simplify reading of the figure. We agree that there is not much more scientific reason behind it. We would like however, if the Editor agrees, to keep the separated illustration, as we agree with the previous reviewer that showing the pathway at once makes it very difficult to dissect the PPP and glycolytic reactions. We improved the figure legend to make clearer that the pathways are only separated for illustrative reasons.

2. There are a number of metabolites whose non-enzymatic formation has been reported, e.g. ribose-2,4-bisphosphate, hexose-2,4,6-triphosphate (Mueller et al., 1990; Helvetica Chimica Acta 73:1410-1468) or pyrimidine ribonucleotides (Powner et al., 2009; PMID 19444213). In particular the former constitute non-typical metabolites which are in our current understanding not part of modern metabolic systems. Is it possible to test any of these compounds for their degradation products to check whether they are eventually converted into glycolysis or PPP metabolites?

We agree that these are interesting experiments proposed. However, as the reviewer state these metabolites are not part of glycolysis, the pentose phosphate pathway- and there is not an enzyme known that binds them, nor is it clear whether they were associated with the evolution of the metabolic network, nor whether these metabolites really existed in the primordial ocean. Therefore we consider the question whether they are interconverted by a primordial plausible catalyst out of focus of this study.

3. The authors conclude that some metal ions, in particular Fe(II), act as catalysts in the conversion reactions (thereby increasing their specificity). Can the authors exclude that Fe(II) is not a catalyst, but is rather being consumed during the conversion reactions in an unknown mechanism (since Fe(II) is present at higher concentrations (200 μ M vs. 7.5/100 μ M) than the metabolites)? For example, do the conversion rates depend on the concentration of Fe(II), which would indeed suggest that it acts as a catalyst?

We can fully exclude that the products measured contain incorporated iron – this would create a significant mass shift: iron has a standard atomic weight of 55.84, but the SRM selection window is set to an m/z of 0.7. An iron containing molecule would thus have a significant different mass compared to the resolution of the SRM method. Therefore we can exclude that iron is incorporated into the measured products.

However, the other possibility is that Fe(II) could act as co-substrate functioning as electron donor. This is very likely, as iron was not limiting in the Archean ocean, and the redox functions of iron are very well documented (Wehrli, Sulzberger, & Stumm, 1989). We included this possibility in the discussion.

4. Page 20 lines 9-16: the authors discuss the apparent paradox that thermophile organisms can thrive at temperatures which exceed their metabolite's stability. It is not clear to me why this is necessarily a paradox: the turnover rates of metabolites in glycolysis are in the order of seconds for model organisms such as E.coli. Even if one assumes turnover rates which are 1-2 orders of magnitude lower, these would still be much higher than the degradation rates reported here, suggesting that metabolite degradation is negligible compared to enzymatic conversion, at least in actively metabolizing cells. The authors should make this aspect clear in the discussion.

In this statement we try to show that the critical point is not about the reaction rate, but about the specificity: Metabolism uses only a subset of the chemical possible molecules; if metabolites would randomly react into non metabolic intermediates, the network topology would get distorted. We thank the reviewer that this was not sufficiently clear – in order to avoid confusion of the reader, this discussion point has been fully removed, and will be presented in more detail in a future paper.

5. Some of the conversion rates are in the nano-molar range - how did the authors decide whether a particular metabolite is formed at such slow rates vs not formed at all? Did they use a metabolite-specific detection limit, or a general concentration cut-off?

We determined the limit of detection for every metabolite separately by using the standard procedures of analytical calibration curves. A typical limit of detection for an SRM method on our setup (Agilent 6460 + Agilent 1290) in ~ femtomol/OC (for many metabolites even below), a nanomol value is thus already ~1000x about the LOD. For quantification, a 15x S/N ratio was considered the minimum. The materials and methods section has been improved for clarity.

TRANSFERRED REVIEW #2

This article presents some surprising and unexpected results about the stability and spontaneous chemical conversions of intermediary metabolites that has relevance to the evolution of metabolism. Though one might worry about the presence of traces of enzymes remaining in the metabolite samples, the authors have excluded this as a plausible explanation of the results.

Major point

1. There is a weakness in the argument on p. 19 that these results could account for the formation of ribose 5P, since net formation of pentose sugars has only been shown to occur from 6-phosphogluconate. Now there is no interconversion shown that generates 6-phosphogluconate. Indeed, the authors themselves argue that the oxidative pentose phosphate pathway is not universal in archaea and could have arisen later. Organisms that lack the oxidative pentose pathway can form

ribose 5P from hexoses and trioses, but the necessary conversions are not present in their demonstrated reaction set. This inconsistency in the argument should certainly be removed, but in my view, they have not provided any new support for this weak point of the RNA world hypothesis. We agree with the reviewer, and this discussion point has been strongly revised to not overstate our results. What we left in the discussion is the statement that sugar phosphate interconversions sequences should be considered as a prebiotic source for ribose 5-phosphate, and mention that ribose 5-phosphate was more stable than the other pentose phosphates that we tested.

Minor points

2. Sentence at lines 17-19, p. 8. I was unclear what was implied by this sentence, since it seems to contradict the following sentence. Do they mean they saw no significant differences unless they included the metal ions listed? If so, that would be a shorter and clearer statement.

3. The text needs significant sub-editing to remove disagreements between subjects and verbs, changes of tense within sentences and typographic errors.

4. The reference for Fuchs (2011) is incomplete.

We revised the text accordingly.

References

- Amend, J. P., LaRowe, D. E., McCollom, T. M., & Shock, E. L. (2013). The energetics of organic synthesis inside and outside the cell. *Philosophical Transactions of the Royal Society of London. Series B, Biological Sciences*, 368(1622), 20120255. doi:10.1098/rstb.2012.0255
- Braakman, R., & Smith, E. (2012). The emergence and early evolution of biological carbon-fixation. *PLoS Computational Biology*, 8(4), e1002455. doi:10.1371/journal.pcbi.1002455
- Clasquin, M. F., Melamud, E., Singer, A., Gooding, J. R., Xu, X., Dong, A., ... Caudy, A. A. (2011). Riboneogenesis in yeast. *Cell*, 145(6), 969–980. Retrieved from http://www.ncbi.nlm.nih.gov/entrez/query.fcgi?cmd=Retrieve&db=PubMed&dopt=Citation&list_uids=21663798
- Fraenkel, D. G. (1986). Mutants in glucose metabolism. *Annu Rev Biochem*, 55, 317–337. Retrieved from http://www.ncbi.nlm.nih.gov/entrez/query.fcgi?cmd=Retrieve&db=PubMed&dopt=Citation&list_uids=3017192
- Fuchs, G. (2011). *Alternative pathways of carbon dioxide fixation: insights into the early evolution of life? Annual review of microbiology* (Vol. 65, pp. 631–58). doi:10.1146/annurev-micro-090110-102801
- Itoh, A., Ohashi, Y., Soga, T., Mori, H., Nishioka, T., & Tomita, M. (2004). Application of capillary electrophoresis-mass spectrometry to synthetic in vitro glycolysis studies. *Electrophoresis*, 25(13), 1996–2002. doi:10.1002/elps.200305905
- Nunoura, T., Takaki, Y., Kakuta, J., Nishi, S., Sugahara, J., Kazama, H., ... Takami, H. (2011). Insights into the evolution of Archaea and eukaryotic protein modifier systems revealed by the genome of a novel archaeal group. *Nucleic Acids Research*, 39(8), 3204–23. doi:10.1093/nar/gkq1228
- Szostak, J. W., Bartel, D. P., & Luisi, P. L. (2001). Synthesizing life. *Nature*, 409(6818), 387–90. doi:10.1038/35053176
- Wehrli, B., Sulzberger, B., & Stumm, W. (1989). Redox processes catalyzed by hydrous oxide surfaces. *Chemical Geology*, 78(3-4), 167–179. doi:10.1016/0009-2541(89)90056-9

2nd Editorial Decision

10 March 2014

Thank you again for submitting your revised work to Molecular Systems Biology. We are now globally satisfied with the modifications made and we are pleased to inform you that we will be able to accept your manuscript for publication, pending the following minor points:

Please find attached the Word file with some small suggestions for edits.

2nd Revision - authors' response

11 March 2014

We are very pleased about the acceptance of our manuscript for publication in MSB, and look now really forward to the publication process.

We are thankful to the Editor for the suggestions in the manuscript text, and included them into the final document.